# Regioselective fluorination of allenes enabled by I(I)/I(III) catalysis

Zi-Xuan Wang[1], Yameng Xu[1] & Ryan Gilmour [1]✉

The prominence and versatility of propargylic fluorides in medicinal chemistry, coupled with the potency of F/H and F/OH bioisosterism, has created a powerful impetus to develop efficient methods to facilitate their construction. Motivated by the well-established conversion of propargylic alcohols to allenes, an operationally simple, organocatalysis-based strategy to process these abundant unsaturated precursors to propargylic fluorides would be highly enabling: this would consolidate the bioisosteric relationship that connects propargylic alcohols and fluorides. Herein, we describe a highly regioselective fluorination of unactivated allenes based on I(I)/I(III) catalysis in the presence of an inexpensive HF source that serves a dual role as both nucleophile and Brønsted acid activator. This strategy enables a variety of secondary and tertiary propargylic fluorides to be prepared: these motifs are prevalent across the bioactive small molecule spectrum. Facile product derivatisation, concise synthesis of multi-vicinal fluorinated products together with preliminary validation of enantioselective catalysis are disclosed. The expansive potential of this platform is also demonstrated through the highly regioselective organocatalytic oxidation, chlorination and arylation of allenes. It is envisaged that the transformation will find application in molecular design and accelerate the exploration of organofluorine chemical space.

In the suite of molecular editing strategies that are routinely leveraged in contemporary drug discovery, fluorination is privileged[1,2]. The subtle bioisosteric exchange of H or OH to F introduces a powerful physicochemical modulator that enables key parameters to be precisely tailored with negligible steric consequences[3–8]. In the case of hydroxyl to fluorine substitution, the elimination of specific hydrogen bonds can be achieved whilst preserving the local electronic and steric environment: this frequently elicits striking changes on $pK_a$, lipophilicity, and metabolic stability[9–13]. The juxtaposition of the small van der Waals radius of fluorine (1.47 Å)[3] and its high electronegativity also manifests itself in an array of fluorine-specific conformational effects[14,15]: these are inextricably linked to the low-lying antibonding orbital of the C–F bond ($\sigma^*_{C–F}$). It logically follows that the introduction of C(sp³)–F bonds proximal to π-systems permits a range of stabilizing non-covalent interactions to operate, and this often gives rise to structural preorganization in a manner complementary to steric control[16]. Propargylic

fluorides are the simplest exemplar of this structural group, and their prevalence across the pharmaceutical and agrochemical landscapes is conspicuous[17,18]. In particular, the success of secondary (2°) and tertiary (3°) propargylic fluoride motifs[19,20] (Fig. 1A) is a persuasive argument for continued innovation in the development of effective synthetic methods to enable construction[21–28].

Motivated by the arsenal of synthetic methods to convert propargylic alcohols to allenes[29,30], it was envisaged that a strategy to access the corresponding propargylic fluorides from these abundant starting materials would strengthen the bioisosteric relationship between two important drug discovery modules (Fig. 1B). Although deceptively unassuming, the title transformation presents a series of historically persistent regioselectivity challenges that must be overcome. Skeletal rearrangements[31,32], and the formation of isomeric mixtures (branched versus linear)[33] continue to complicate the development of a regioselective I(I)/I(III)-catalyzed fluorination of

[1]Institute for Organic Chemistry, University of Münster, Corrensstraße 36, 48149 Münster, Germany. ✉e-mail: ryan.gilmour@uni-muenster.de

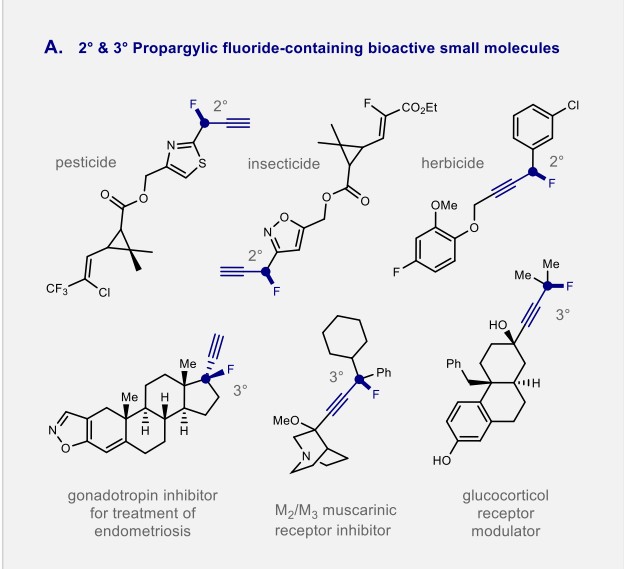

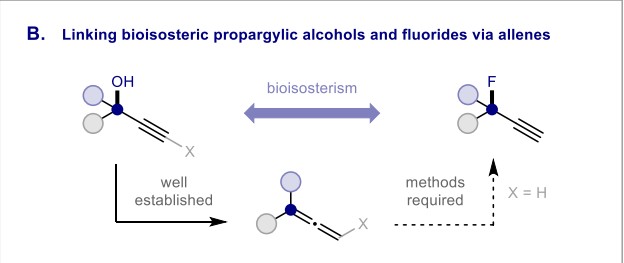

**Fig. 1 | Motivation and conceptual framework for the study. A** Selected examples of bioactive small molecules that contain secondary and tertiary propargylic fluoride units. **B** Bioisosteric relationship of propargylic fluorides and alcohols. **C** Stoichiometric *geminal* difluorination of allenes. **D** Reaction blueprint to enable catalysis-based regioselective fluorination of allenes.

allenes. However, the ability of I(III)-species to activate allenes towards fluorination provides an important foundation for reaction development. Elegant studies by Murphy and co-workers have established that stoichiometric hypervalent iodine reagents, working in concert with a boron trifluoride etherate Lewis acid activator, allow phenyl-substituted allenes to undergo a difluorination / phenonium ion rearrangement sequence to liberate α-(difluoromethyl)styrenes (Fig. 1C)[31]. In contrast to this Lewis acid-based stoichiometric process to liberate geminally difluorinated products, it was envisaged that a catalytic paradigm operating under Brønsted acid activation might enable divergent selectivity to be achieved (Fig. 1D). Cognisant of the aptitude of I(I)/I(III) catalysis to orchestrate the selective fluorination of short π-systems[34–41], a conceptual platform was developed that hinged on the in situ formation of an ArIF$_2$ species by chemical oxidation[42,43]. In a reversal of circumstances, it was reasoned that the I(III) species would selectivity engage the terminal unit of the allene through ligand exchange (**I**). Formation of the C(sp$^3$)–F bond would then generate an alkenyl iodonium species (**II**) that would be susceptible to elimination to liberate the product and regenerate the ArI catalyst. This would constitute an operationally simple approach to generate 2° and 3° propargylic fluorides and address a long-standing regioselectivity challenge (branched versus linear) in the I(III)-mediated fluoro-functionalization of allenes.

Herein we report a highly regioselective fluorination of unactivated allenes based on I(I)/I(III) catalysis to generate variety of secondary and tertiary propargylic fluorides. By merging this process with subsequent downstream manipulations, it has been possible to create a broad platform from which to expand the organofluorine chemical space.

## Results and discussion

### Reaction development

To establish the viability of a regioselective, organocatalytic fluorination platform, allene **S1** was selected as a model substrate for reaction development. This aliphatic derivative was chosen to eliminate the possibility of phenonium ion rearrangement chemistry. A screen of inexpensive aryl iodide catalysts was then conducted in the presence of an amine•HF complex (1:5.0) and Selectfluor® as the terminal oxidant (Fig. 2A). This revealed 1-iodo-2,4-dimethylbenzene to be a highly effective catalyst for the title transformation, furnishing the desired propargylic fluoride **1** in 78% yield. Importantly, the reaction proved to be highly regioselective with the desired branched product being preferentially formed over the linear product (>20:1). It is important to note that 1,2-difluorination was not observed under the reaction conditions.

Reactions performed in the absence of the catalyst, oxidant, and amine•HF complex were not productive (entries 1–3), which supports the involvement of an I(I)/I(III) cycle. Modifying the oxidant (entry 4) or the amine:HF ratio (entries 5 and 6, vide infra), proved to be detrimental to efficiency. Lowering the catalyst loading to 20 mol% (entry 7) had a minor impact on efficiency, whereas reactions performed at 10 mol% (entry 8) had a more noticeable effect.

To further demonstrate the influence of Brønsted acidity on the reaction outcome[44], the amine:HF ratio was plotted against the yield of fluorination as determined by $^{19}$F NMR spectroscopy (Fig. 2B). This revealed a plateau with the 1:5 amine:HF ratio, and is highly reminiscent of the role of TFA in chlorination reactions involving the venerable Willgerodt Reagent (PhICl$_2$)[45,46].

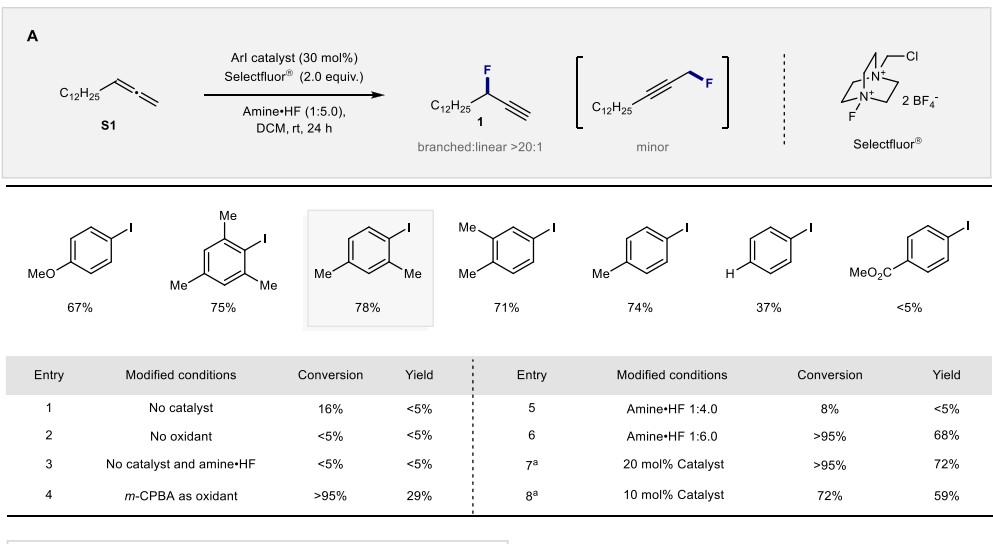

| Entry | Modified conditions | Conversion | Yield | Entry | Modified conditions | Conversion | Yield |
|---|---|---|---|---|---|---|---|
| 1 | No catalyst | 16% | <5% | 5 | Amine•HF 1:4.0 | 8% | <5% |
| 2 | No oxidant | <5% | <5% | 6 | Amine•HF 1:6.0 | >95% | 68% |
| 3 | No catalyst and amine•HF | <5% | <5% | 7[a] | 20 mol% Catalyst | >95% | 72% |
| 4 | m-CPBA as oxidant | >95% | 29% | 8[a] | 10 mol% Catalyst | 72% | 59% |

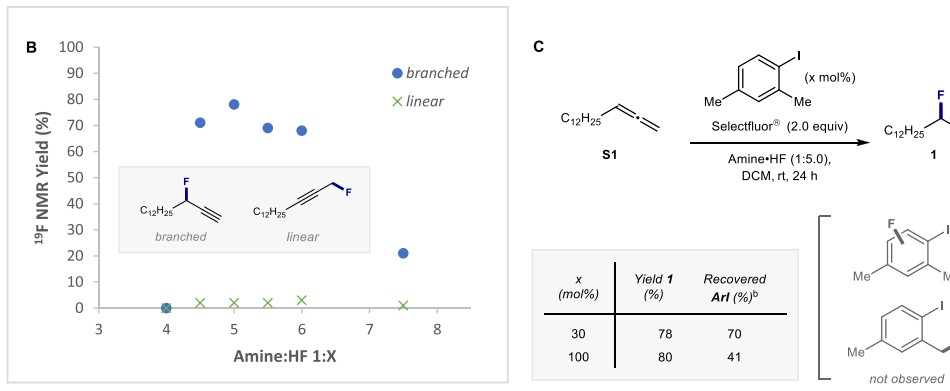

**Fig. 2 | Reaction development. A** Optimization of reaction conditions. Standard reaction conditions: allene **S1** (0.1 mmol), catalyst (30 mol%), amine•HF 1:5.0 (0.25 mL), DCM (0.25 mL), and Selectfluor® (0.20 mmol). Ethyl fluoroacetate was used as the internal standard. Conversions were determined by ¹H NMR. Yields were determined by ¹⁹F NMR. The regioselectivity (branched versus linear) was >20:1 in all cases. **B** Influence of the amine:HF ratio on the transformation. **C** Control experiments. [a] 1.5 equiv. Selectfluor® was used. [b] Yields were determined by ¹H NMR. The amine·HF mixtures were prepared based on commercially available NEt₃·3HF and Py·9.23HF (Olah's reagent). DCM dichloromethane, mCPBA meta-chloroperoxybenzoic acid.

Since electron-rich iodobenzene derivatives pose the risk of reacting directly with Selectfluor®[47], a control experiment with stoichiometric quantities of the catalyst (1-iodo-2,4-dimethylbenzene) was performed (Fig. 2C). This enabled the propargylic fluoride **1** to be generated in 80% yield, and 1-iodo-2,4-dimethylbenzene to be recovered in 41% yield. No fluorination of the catalyst was observed in the crude reaction mixture or post-reaction. Repeating this analysis on the catalytic reaction revealed closely similar results, with 70% of 1-iodo-2,4-dimethylbenzene recovered (determined by ¹H NMR). Based on these preliminary data, the involvement of a fluorinated 1-iodo-2,4-dimethylbenzene derivative in the catalytic cycle can be discounted.

## Substrate scope

Having established suitable reaction conditions for the regioselective fluorination of allenes, the scope and limitations of the transformation were explored. Since the organocatalyst is commercially available and inexpensive, a catalyst loading of 30 mol% was employed to maximize product formation. Subtle variations in the amine:HF ratio were also found to modulate reaction efficiency, which is fully in line with our previous observations pertaining to the influence of Brønsted acidity in I(I)/I(III)-catalyzed fluorinations[39–41]. To that end, representative substrates were exposed to four amine:HF ratios ranging from 1:4.5 through to 1:6.0, and the most effective was reported (Fig. 3, Methods A to D). To differentiate this study from previous work on activated allenes, various alkyl allenes were initially evaluated. Gratifyingly, simple alkyl and cycloalkyl derivatives were compatible with the

reaction conditions, as exemplified by fluorides **1**, **2**, and **12**. Functional groups including halides (**3**), ethers (**4**), alcohols (**6**), tosylates (**7**), and phthalimides (**8**) were all tolerated (up to 77% yield), and the presence of potentially sensitive benzylic positions also proved to be unproblematic (**5**).

Pleasingly, medicinally relevant heterocycles such as pyridines (**9**) and thiazoles (**10**) were also compatible with the reaction conditions (up to 82% yield). Moreover, the transformation proved to be highly chemoselective for the allene motif in substrates containing cyclopropanes (**11**) or cinnamoyl motifs (**15**). Allenes derived from estrone (**13**) and ibuprofen (**14**) were smoothly processed to branched propargylic fluorides in good yields and regioselectivities (>20:1), demonstrating the synthetic utility of the method in more complex settings.

To extend the scope of the method further, aryl allenes were then examined as substrates. Under stoichiometric conditions with a Lewis acid activator, these substrates are known to undergo phenonium ion rearrangements to generate α-(difluoromethyl)styrenes (Fig. 1C)[31]. However, this catalytic protocol with Brønsted acid activation switched the selectivity to enable a variety of electron-deficient propargylic fluorides with varying substitution patterns to be generated in a highly regioselective manner (up to 17:1, branched:linear). This is exemplified by the trifluoromethyl derivatives **16** and **17**, ester **18**, amide **19**, and the methylsulfonyl derivative **21**. Gratifyingly, aryl triflates were also compatible with the reaction conditions, which provides a handle for subsequent downstream synthetic manipulations

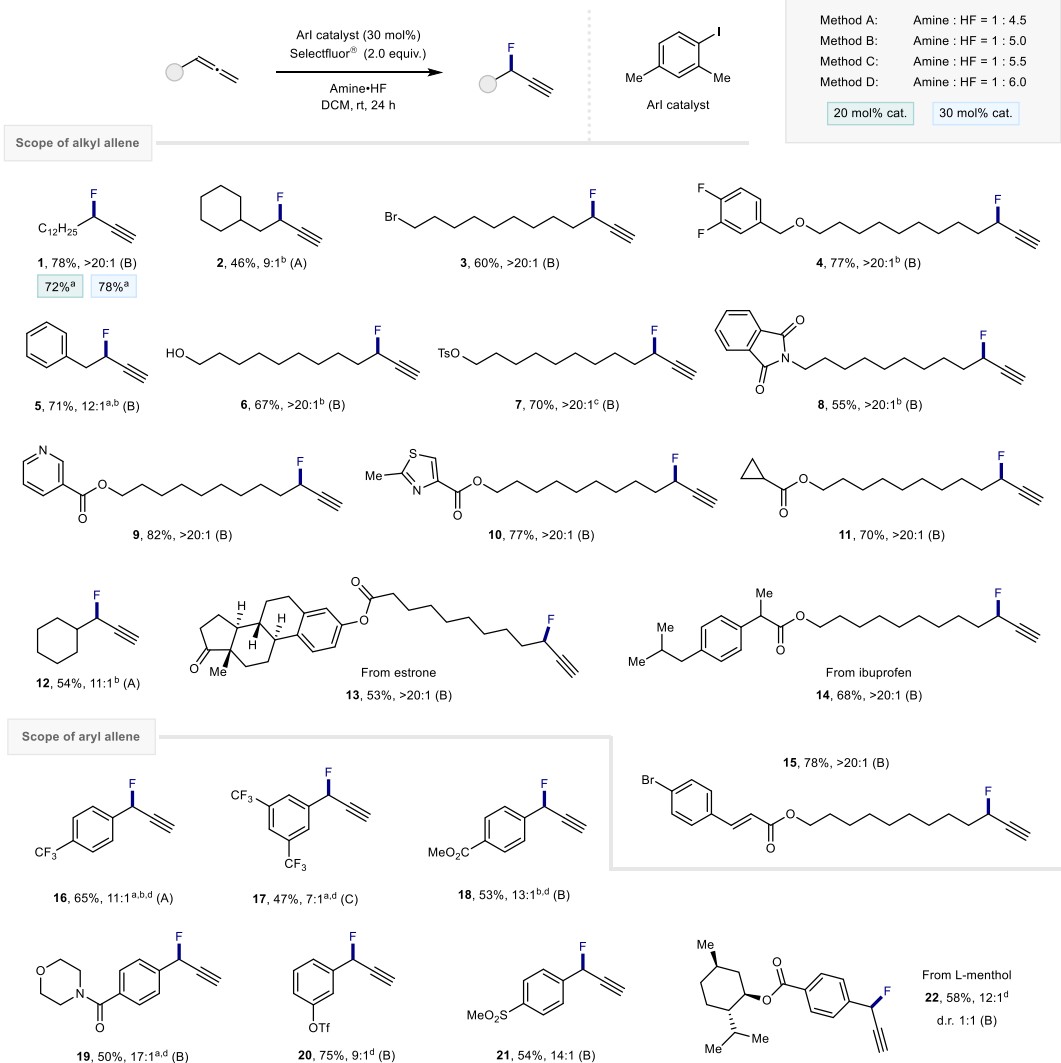

**Fig. 3 | Exploring the scope of the title transformation.** Reaction conditions: allene (0.1 mmol), catalyst (30 mol%), amine•HF (0.25 mL), DCM (0.25 mL) and Selectfluor® (0.2 mmol). Isolated yields reported. The regioselectivity was determined by [19]F NMR from the crude reaction mixture. [a] [19]F NMR yield using ethyl fluoroacetate as the internal standard. [b] Reaction performed on 0.20 mmol scale. [c] Reaction performed on 0.15 mmol scale. [d] DCE was used as solvent. The enantiomer of the product shown was arbitrarily shown.

(**20**). Finally, the regioselective fluorination of the L-menthol substituted aryl allene was also achieved to access compound **22**.

To enable the construction of tertiary propargylic fluorides, attention was turned to exploring the regioselective fluorination of 1,1-disubstituted allenes (Fig. 4). A range of α-chloroalkylated allenes with aryl and alkyl substituents were initially subjected to the general catalysis conditions, and this led to the formation of tertiary propargylic fluorides (up to 78%) which contain alkyne and C(sp³)-Cl handles for facile derivatization (**23**–**26**). It is interesting to note that this method enables products, such as **23** and **24**, to be forged which contain C(sp), C(sp²), and C(sp³) substituent directly bound to the fluorine-bearing center.

Moreover, the tertiary propargylic fluoride **23** could be generated with only a slight reduction in yield using 20 mol% catalyst loading. Replacing CH₂Cl by CH₂F as the allene substituent was also successful, enabling a *vicinal* difluoride motif (**27**) to be generated in 66% yield[15]. Given the importance of homopropargyl ether and amine fragments in the synthesis of bioactive molecules, α-oxy and aminoalkylated allenes were introduced [**28** (61%) and **29** (55%)], in addition to the unsubstituted derivative **30**. The compatibility of the method with a D-menthol derived allene was also validated (**31**, 73%).

## Synthetic applications

Telescoping the reaction from 0.1 mmol to 1.5 and 2.0 mmol scale was then demonstrated with compounds **23** and **26**, respectively (Fig. 5A). These experiments illustrate the synthetic utility of this approach in generating versatile fluorinated linchpins. To further demonstrate this, alkyne **23** was converted to aryl and acyl substituted internal alkynes **32** and **33** through Pd-catalyzed Sonogashira coupling (Fig. 5B). Complete and semi-hydrogenation of alkyne **1** proved facile, affording alkane and alkene products **34** and **35**, respectively. Ag-catalyzed oxidation of alkyne **1** enabled the α-fluoro ketone **36** to be prepared in an operationally simple manner[48]. Finally, triazole **37** was generated through a copper-catalyzed click reaction, further underscoring the value of propargylic fluorides in heterocycle formation. To demonstrate the synthetic potential of the transformation in generating multiply fluorinated scaffolds, a sequential I(I)/I(III) catalysis-based protocol was devised (Fig. 5C)[49]. Crabbé reaction of alkyne **26** enabled the allene **38** to be prepared in 56% yield: this species was then subjected to an I(I)/I(III) catalysis cycle, enabling regioselective 1,2-difluorination to afford **39** as the sole product (>20:1 Z/E). Moreover, preliminary validation of enantioselective catalysis was achieved using chiral resorcinol-derived aryl iodides as chiral mediators[50,51]. This

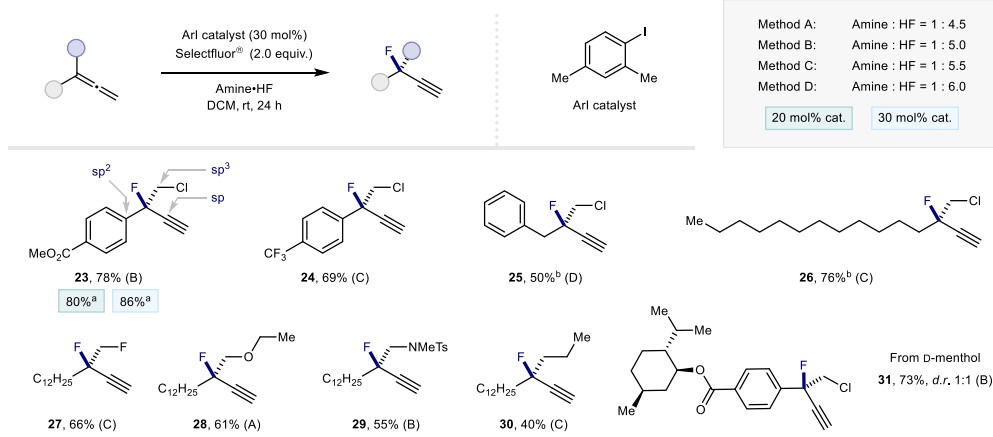

**Fig. 4 | Expanding scope to 1,1-disubstituted allene.** Reaction conditions: allene (0.1 mmol), catalyst (30 mol%), amine•HF (0.25 mL), DCM (0.25 mL) and Select-fluor® (0.2 mmol). Isolated yields reported. [a] $^{19}$F NMR yield using ethyl fluoroacetate as internal standard. [b] Reaction time increased to 48 h. The enantiomer of the product shown was arbitrarily shown.

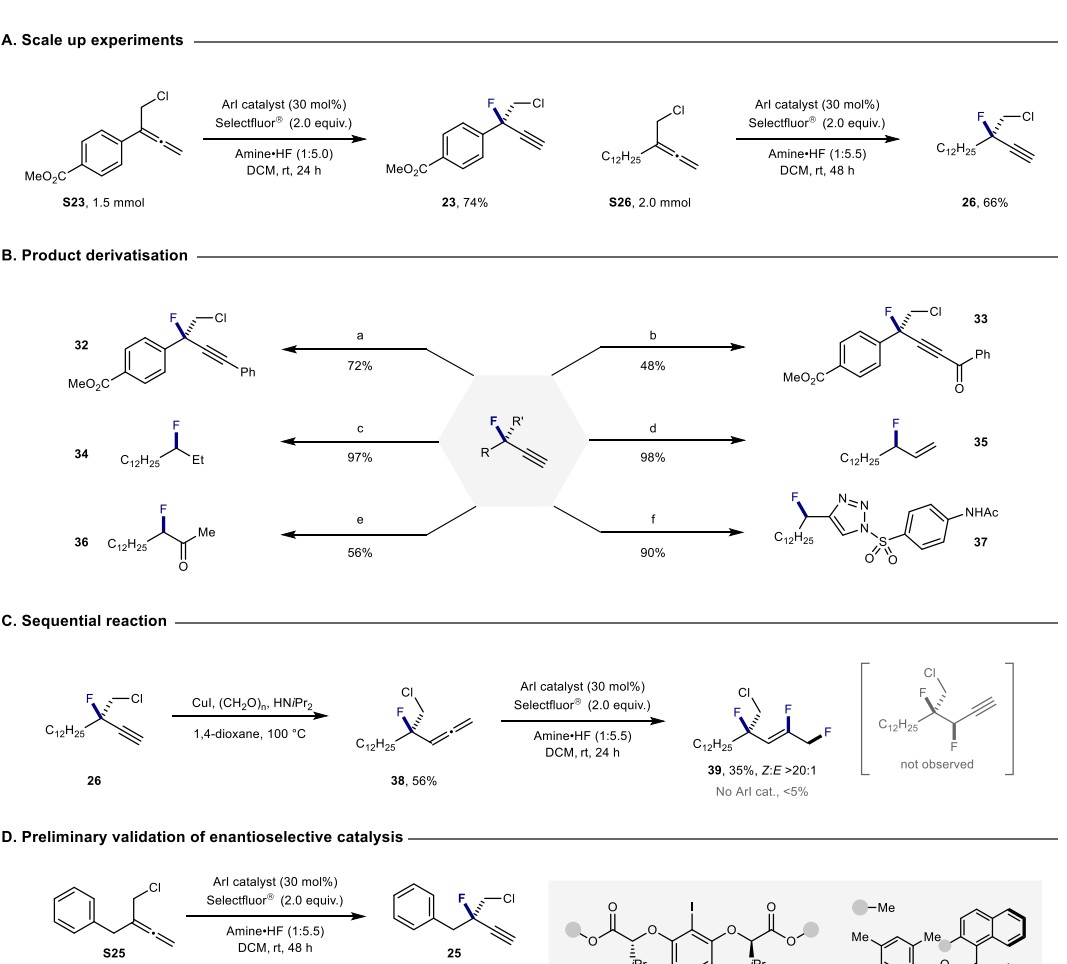

**Fig. 5 | Synthetic applications. A** Scale-up experiments. **B** Product derivatization. Conditions: [a] Pd(PPh$_3$)$_4$ (10 mol%), CuI (10 mol%), iodobenzene (2.0 equiv.), NEt$_3$ (0.33 M), 60 °C, 12 h. [b] Pd(PPh$_3$)$_4$ (10 mol%), CuI (15 mol%), benzoyl chloride (3.0 equiv.), NEt$_3$ (0.25 M), 30 °C, 12 h. [c] Pd/C (10 mol%), H$_2$, MeOH (0.1 M), r.t., 24 h. [d] Lindlar catalyst (20 wt%), quinoline (1.0 equiv.), pyridine (0.17 M), H$_2$, r.t., 30 min.

[e] AgSbF$_6$ (30 mol%), MeOH, H$_2$O (10:1, 0.12 M), 80 °C, 24 h. [f] CuTc (10 mol%), 4-acetamidobenzenesulfonyl azide (1.2 equiv.), toluene (0.2 M), r.t., 12 h.
**C** Sequential reaction. **D** Preliminary validation of enantioselective catalysis. Isolated yields are given. The enantiomer of the product shown was arbitrarily shown.

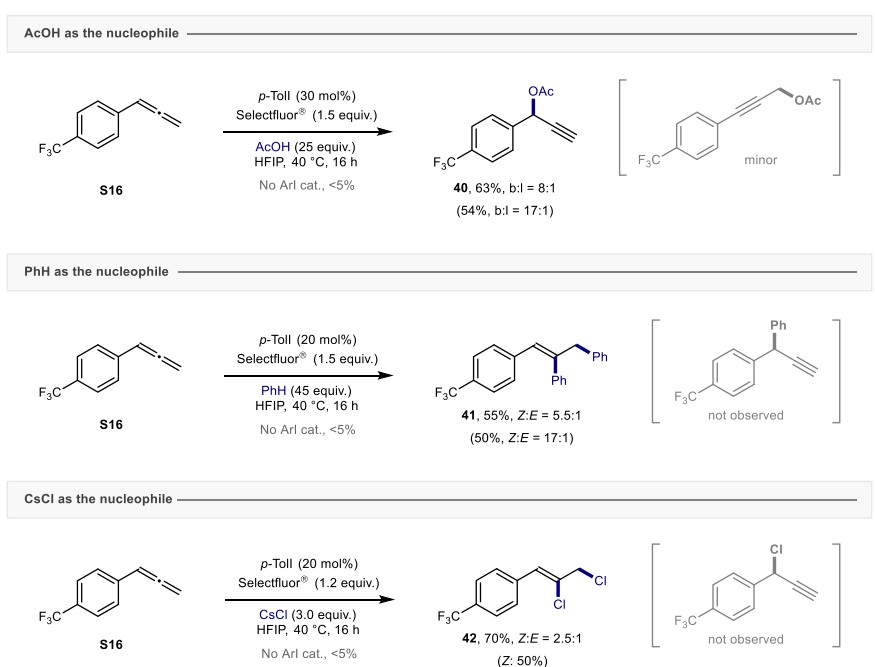

**Fig. 6 | Expanding the scope of the catalytic functionalization of allenes enabled by I(I)/I(III) catalysis.** Combined $^{19}$F NMR yields of both isomers were provided and determined by $^{19}$F NMR using ethyl fluoroacetate as the internal standard. The regioselectivity ratio was determined by $^{19}$F NMR analysis of the crude reaction mixture. Isolated yields are given in parentheses.

enabled the tertiary propargylic fluoride **25** to be obtained in 73.5:26.5 er and 76.5:23.5 er (Fig. 5D). These results constitute an encouraging platform for the future development of enantioselective, organocatalytic allene fluorination protocols.

### General catalytic functionalization of allenes

To explore the expansiveness of this transformation, the aptitude of the catalysis to orchestrate related allene functionalization reactions was investigated. It was envisaged that the introduction of exogenous nucleophiles other than fluoride, which would necessarily be accompanied by variations in Brønsted acidity, may reveal unprecedented structural motifs (Fig. 6). Our initial investigation focused on replacing HF by AcOH to generate propargyl acetates as surrogates of the biosisteric alcohols described in Fig. 1. Under the standard catalysis conditions, the reaction proved ineffective. However, the introduction of hexafluoroisopropanol (HFIP) as the reaction medium proved to be effective in unlocking the desired reactivity[52-54]: this allowed the propargyl acetate **40** to be prepared in good yield and regioselectivity. This further underscores the value of HFIP in I(I)/I(III)-catalyzed acetoxylation reactions. Attempts to synthesize the propargyl amine regioselectively using NHTs$_2$ as the nucleophile were unsuccessful (see Supplementary Information)[33]. Leveraging benzene as a non-protic nucleophilic solvent was then explored. This led to a selectivity switch such that diarylation occurred to yield the highly functionalized styrene **41** (55%, Z:E = 5.5:1). Moreover, a similar strategy utilizing CsCl as the nucleophile enabled the 1,2-dichloride **42** to be generated in 70% yield (Z:E = 2.5:1)[55,56]. In all cases, the removal of the organocatalyst completely suppressed the acetoxylation, arylation, and chlorination of allene **S16**. It is tempting to speculate that these data open up a wealth of possibilities for the regioselective difunctionalization of simple allenes by I(I)/I(III) catalysis.

Inspired by the functional diversity of propargylic fluorides in the small molecule agrochemical and pharmaceutical arena, a highly regioselective fluorination of unactivated allenes has been realized by a metal-free I(I)/I(III) catalysis platform. This blueprint enables readily available allenes to be processed to an array of secondary and tertiary propargylic fluorides with broad functional group tolerance.

By operating under the auspices of I(I)/I(III) catalysis, in which HF serves as a source of nucleophile and Brønsted acid activator, regiochemical challenges and skeletal rearrangements are mitigated, unlocking divergent selectivity that is complementary to that is observed using stoichiometric I(III) reagents. The synthetic utility of the transformation is demonstrated through facile product derivatization, scale-up experiments, and preliminary validation of enantioselective catalysis. Integrating this process in a reaction sequence has also been achieved to generate a multiply fluorinated product with clearly defined sites for yet further modification. The expansive potential of the transformation has been demonstrated in acetoxylation, and the ability to modulate regioselection, are illustrated in diarylation and dichlorination reactions. Further expanding the generality of this organocatalytic functionalization of allenes beyond the construction of C−F, C−O, C−Cl, and C−C bonds will be the subject of future investigations.

## Methods

### General procedure for regioselective fluorination of allenes

Unless otherwise stated, a Teflon® vial was equipped with a 1 cm stirring bar followed by the addition of allene (0.1 mmol, 1.0 eq.), 1-iodo-2,4-dimethylbenzene (7 mg, 0.03 mmol, 30 mol%) and DCM (0.25 mL). The stated amine:HF mixture was added (0.25 mL) via syringe. After stirring for 1 min, Selectfluor® (71 mg, 0.2 mmol, 2.0 eq.) was added in one portion. The reaction vessel was then sealed with a Teflon® screw cap. After stirring (350 rpm) at ambient temperature for 24 h, the reaction mixture was poured into 100 mL of a saturated solution of NaHCO$_3$ (CAUTION, generation of CO$_2$!). The Teflon® vial was rinsed with DCM and dropped into another flask of saturated aqueous solution of NaHCO$_3$ to guarantee the removal of excess HF. The organics were extracted with DCM (3 × 30 mL), the combined organic layers were dried over Na$_2$SO$_4$, filtered and the solvent was carefully removed under reduced pressure. An internal standard (ethyl fluoroacetate) was added to the crude residue. The NMR yield and regioselectivity ratio (branched:linear) were analyzed by $^{19}$F NMR spectroscopy against the internal standard. The NMR sample was recombined with the crude residue and purification by

column chromatography or preparative thin-layer chromatography yielded the desired product.

## Data availability

Supplementary Information is available for this paper. All data are available in the main text or the supplementary materials. Data supporting the findings of this manuscript are also available from the authors upon request.

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

## Acknowledgements
We gratefully acknowledge the support provided by the technical departments of the Institute for Organic Chemistry at the University of Münster. We acknowledge financial support from the European Research Council (ERC Consolidator Grant RECON 818949, R.G.), the China Scholarship Council (Y.X.), and the Alexander von Humboldt Foundation (Z.-X.W.).

## Author contributions
Conceptualization: Z.-X.W., R.G.; Methodology: Z.-X.W., R.G.; Investigation: Z.-X.W., Y.X.; Funding acquisition: Z.-X.W., R.G.; Project administration: Z.-X.W., R.G.; Writing: Z.-X.W., R.G.

## Funding

## Competing interests
The authors declare no competing interests.
