## [Peer Review File · Nature Communications]

REVIEWER COMMENTS

Reviewer #1 (Remarks to the Author):

Gilmour and coworkers demonstrate a method for the fluorination of alkyl and aryl-substituted allenes by I(I)/I(III) catalysis. A series of secondary and tertiary propargylic fluorides were achieved in moderate to good yields in high regioselectivities. In the transformation, the HF source serves as a dual role (nucleophile and Brønsted acid activator). The authors tried to use chiral ArI catalyst for the reaction. But the desired product was generated in low enantioselectivity. When they extended the similar conditions to the esterification of alkenes using AcOH instead of the HF source, the corresponding product was formed in decent yield. Using benzene and CsCl as the nucleophiles instead of the HF source, 1,2-difunctionalized products were formed with Z/E isomers under the similar conditions. This work is not novel on the basis of the iodine-involved process. However, this method provides a good pathway for the synthesis of secondary and tertiary propargylic fluorides, a class of potentially useful compounds. Another interesting point is that the fluorine system affects the reactivity of the iodine compound. Furthermore, the Supplementary Information is well prepared. For this work, some issues are still needed to be addressed:

(1) In Murphy's work (ref. 31), catalytic amounts of BF₃ were used. If BF₃ (catalytic amounts or stoichiometric) is used instead of the HF source in this work, what will happen? How about using HBF₄?

(2) In this work, terminal allenes are always utilized. How about using non-terminal allenes?

(3) Using an amine source instead of the HF source, such as those in the literature (ref. 33), what will happen? The authors are suggested to mention the results in the text.

(4) Compared with the transformation in the Murphy's work (ref. 31), this transformation gives the desired products without a rearrangement. It seems that the fluorine source affects the reactivity of the reaction. The authors should explain the reason in the text, why the reaction does not go through a rearrangement under the conditions.

(5) In Fig. 1D, intermediate II undergoes hydrogen elimination to give the alkynes. Why is the intermediate not converted to a difluorinated product by the attack of a second fluoride? The authors should explain it in the text.

Reviewer #2 (Remarks to the Author):

The study by Gilmour and colleagues explores the formation of propargylic fluorides from allenes using an aryl iodide-catalyzed process. The fluoride source is an amine.HF complex and Selectfluor is used as the oxidant for the in situ formation of an ArIF₂ species. The reaction exhibits high selectivity towards the branched isomer and can tolerate a variety of functional groups. In total, 20 alkyl allenes and electron-poor aryl allenes have successfully been converted into propargylic fluorides using this protocol. Additionally, it was found that 1,1-disubstituted allenes are also suitable substrates for this transformation. The authors also report on the functionalization of propargylic fluorides, and an enantioselective variant with a chiral aryl iodide as the catalyst was tested. The observed ee of 47% is modest, but I think that this could be improved by finetuning the chiral aryl iodide catalyst.

Furthermore, the authors investigated the possibility of expanding the reaction to nucleophiles other than fluoride. The desired propargylic ester was obtained with AcOH when hexafluoroisopropanol was used as a solvent. However, the reaction with benzene or chloride as nucleophiles led to the formation of styrenes. It remains unclear if this is a general phenomenon, as only one example was reported in both cases.

In summary, the study presents an elegant application of I(I)/I(III) catalysis for the formation of propargylic fluorides, complementing existing protocols such as the formation of propargylic fluorides from allenylsilanes. I recommend that the manuscript be published in Nature Communications once the following points have been addressed:

From the mechanistic proposal shown in Figure 1D, it appears that the Selectfluor is required to oxidize the ArI to an ArIF₂ as described in the report by Shreeve and co-workers (reference 43). However, it remains unclear to me how the intermediate I is formed and why an excess of HF is required for the reaction to proceed. I suspect that the authors assume an initial protonation of the ArIF₂ species by the amine HF to yield a cationic ArIF⁺, which then raises the question of whether the reaction also proceeds with a Bronsted acid other than HF. The authors should explain the role of the amine-HF complex in more detail in the manuscript. In this sense, the authors should also provide information about the nature of the amine-HF complex in the manuscript and not only in the Supporting Information.

I also suggest that the authors provide an explanation for why alpha-branched alkylallenes (such as commercially available cyclohexallene) were not included in the substrate scope.

Minor: Supporting Information, page 27: Should the sentence: "This compound was prepared in a previous report from our research group." read as "This compound was prepared according to a previous report from our research group."?

Reviewer #3 (Remarks to the Author):

The manuscript by Gilmour et al. entitled "Regioselective Fluorination of Allenes Enabled by I(I)/I(III) Catalysis" describes a strategy for constructing a variety of secondary and tertiary propargylic fluorides, which are prevalent across the bioactive small molecule spectrum. The main premise of the manuscript is the regioselectivity problem in allene fluorination using Selectfluor being solved using this method. The reaction developed is indeed interesting and displays broad scope. At the end, authors also demonstrate the use of other nucleophiles leading to substituted products. In general, the reaction of Selectfluor with allenes in the presence of nucleophiles has been reported in the literature, leading to different motifs. As such, this method does provide a convenient access to propargylic fluorides but the conceptual novelty is not enough to justify publication in Nature Communications, especially given the absence of a successful enantioselective version (only one example is shown with low enantioselection). While this work is a nice twist to allene-Selectfluor reactivity, it falls short of the expectations of an article to be published in this journal. Addressing the following points may help enhance the manuscript.

1. Is there an explanation of why the yields were low in the case of aryl allenes (Fig. 3).
2. What is the role of cyclopropyl (11) or cinnamoyl motifs (14) for the high chemoselectivity?
3. In the case of tertiary propargylic fluorides, authors have shown limited scope. The late-stage functionalization of important bioactive molecules should also be included, as shown in Fig. 3 (12-14, 21).
4. Do the authors observe the 1,2 difluorination products (like 37) in other cases also? If not, what is the rationale for that?
6. A detailed mechanistic analysis, including computational studies.

•

• **Reviewer #1 (Remarks to the Author):**

Gilmour and coworkers demonstrate a method for the fluorination of alkyl and aryl-substituted allenes by I(I)/I(III) catalysis. A series of secondary and tertiary propargylic fluorides were achieved in moderate to good yields in high regioselectivities. In the transformation, the HF source serves as a dual role (nucleophile and Brønsted acid activator). The authors tried to use chiral ArI catalyst for the reaction. But the desired product was generated in low enantioselectivity. When they extended the similar conditions to the esterification of alkenes using AcOH instead of the HF source, the corresponding product was formed in decent yield. Using benzene and CsCl as the nucleophiles instead of the HF source, 1,2-difunctionalized products were formed with Z/E isomers under the similar conditions. This work is not novel on the basis of the iodine-involved process. However, this method provides a good pathway for the synthesis of secondary and tertiary propargylic fluorides, a class of potentially useful compounds. Another interesting point is that the fluorine system affects the reactivity of the iodine compound. Furthermore, the Supplementary Information is well prepared. For this work, some issues are still needed to be addressed:

Author response: We appreciate the supportive comments from Referee #1 and the insightful suggestions on how to further improve the manuscript.

(1) In Murphy's work (ref. 31), catalytic amounts of BF₃ were used. If BF₃ (catalytic amounts or stoichiometric) is used instead of the HF source in this work, what will happen? How about using HBF₄?

Author response: The referee is quite correct that in the Murphy study (reference 31), the authors used Lewis acid activation of the stoichiometric reagent to generate the *geminal* difluoride. Following the referee's suggestion, the reaction was attempted with both BF₃•OEt₂ and HBF₄. We did not observe formation of the propargylic fluoride (by ¹⁹F NMR), or any other product for that matter, but it was possible to recover some of the starting material after the specified reaction time of 24 h. The crude ¹⁹F NMR spectrums are shown below. We believe that this further demonstrates the importance of Brønsted acid activation (the HF reagent) in these catalytic transformations involving ArIF₂ generation *in situ*.

Entry	Fluoride source	Conversion	Fluorinated products
1	$BF_3 \cdot OEt_2$	14%	NP
2	HBf_4	<5%	NP

(2) In this work, terminal allenes are always utilized. How about using non-terminal allenes?

Author response: This is an excellent suggestion but we envisaged that the terminal alkyne would be advantageous in-line with the preference for terminal alkene in difluorination via I(I)/I(III) catalysis. To test this, we explored the reaction of internal allenes containing ester, alkyl and aryl substituents (please see below).

The reaction of the the allene by ^{19}F NMR spectroscopy revealed that no fluorinated products were generated. However, it was possible to identify remaining starting material after the specified reaction time of 24 h.

In addition, we prepared the alkyl-substituted allene shown below and exposed it to catalysis conditions with various amine:HF ratios. The expected propargylic fluoride was generated, but with extremely poor efficiency.

Finally, the reactions of a model aryl-substituted allene was performed. However, the formation of complex mixtures was observed. The propargylic fluoride and *geminal* difluoride were observed by ^{19}F NMR but with low yields. Purification by column chromatography allowed these compounds to be isolated as an inseparable mixture together with some unidentified impurities.

Reaction of aryl-substituted allene

Entry	Amine : HF	Conversion	¹⁹ F NMR Yield of A	¹⁹ F NMR Yield of B
1	1 : 4.0	63%	NP	NP
2	1 : 4.5	85%	11%	NP
3	1 : 7.5	76%	12%	4%
4	1 : 9.2	76%	12%	5%

Although the current catalytic system is currently limited to terminal allenes, the resulting terminal alkyne provides a useful handle that can be derivatized (**Fig. 5B**, compound **32** and **33**), providing an alternative route to obtain the product from the fluorination with non-terminal allenes. We hope that this addresses the reviewer's comment.

(3) Using an amine source instead of the HF source, such as those in the literature (ref. 33), what will happen? The authors are suggested to mention the results in the text.

Author response: Following the referee's suggestion, we performed the reaction with NHTs₂ as the nucleophile. This led to the formation of the branched (10% by ¹⁹F NMR) and linear (8% by ¹⁹F NMR) propargylic amines. However, significant substrate degradation was observed. This result has been added to the *Supplementary Information* and a statement has been included in the manuscript.

(4) Compared with the transformation in the Murphy's work (ref. 31), this transformation gives the desired products without a rearrangement. It seems that the fluorine source affects the reactivity of the reaction. The authors should explain the reason in the text, why the reaction does not go through a rearrangement under the conditions.

Author response: We thank the referee for this very insightful comment. The work by Murphy and colleagues describes a stoichiometric reaction with the pre-formed *p*-TollF₂ reagent and utilizes a Lewis acid activator (BF₃•OEt₂). In contrast, we report *in situ* generation of *p*-TollF₂ and activation through the intervention of a Brønsted acid. Although we currently do not have structural evidence to support this hypothesis, it is highly likely that the nature of the bound alkenyl iodine intermediates is very different in the two reactions. As reported in citation 44 (*J. Org. Chem.* **82**, 11792 (2017)), we are also aware of fluxional behavior between the ArIF₂ species and the ArIF⁺ species in which the reduced Selectfluor[®] core engages with the cation. This cannot happen in the Murphy reaction (ref. 31) and so it is difficult to draw comparisons between the conditions. Moreover, our reaction conditions introduce a massive excess of the fluoride nucleophile. Collectively, this creates multiple parameters (stoichiometric versus catalytic, oxidant, Lewis versus Brønsted acidity, excess of fluoride nucleophile) that influence the regioselectivities of the two studies. Although we are attempting to gain insights into the intermediates involved, this is incredibly challenging we are unable to clearly attribute the regioselectivity to a specific parameter.

(5) In Fig. 1D, intermediate II undergoes hydrogen elimination to give the alkynes. Why is the intermediate not converted to a difluorinated product by the attack of a second fluoride? The authors should explain it in the text.

Author response: Thank you for this comment. The possibility of S_NV displacement to generate the 1,2-difluoride was never observed and it is likely that the sterically congested environment suppresses this reactivity. We have added a comment to the manuscript indicating that this is not observed.

• **Reviewer #2 (Remarks to the Author):**

The study by Gilmour and colleagues explores the formation of propargylic fluorides from allenes using an aryl iodide-catalyzed process. The fluoride source is an amine.HF complex and Selectfluor is used as the oxidant for the in situ formation of an $ArIF_2$ species. The reaction exhibits high selectivity towards the branched isomer and can tolerate a variety of functional groups. In total, 20 alkyl allenes and electron-poor aryl allenes have successfully been converted into propargylic fluorides using this protocol. Additionally, it was found that 1,1-disubstituted allenes are also suitable substrates for this transformation. The authors also report on the functionalization of propargylic fluorides, and an enantioselective variant with a chiral aryl iodide as the catalyst was tested. The observed ee of 47% is modest, but I think that this could be improved by finetuning the chiral aryl iodide catalyst. Furthermore, the authors investigated the possibility of expanding the reaction to nucleophiles other than fluoride. The desired propargylic ester was obtained with AcOH when hexafluoroisopropanol was used as a solvent. However, the reaction with benzene or chloride as nucleophiles led to the formation of styrenes. It remains unclear if this is a general phenomenon, as only one example was reported in both cases. In summary, the study presents an elegant application of I(I)/I(III) catalysis for the formation of propargylic fluorides, complementing existing protocols such as the formation of propargylic fluorides from allenylsilanes. I recommend that the manuscript be published in Nature Communications once the following points have been addressed:

Author response: We are most grateful to the referee for this very generous summary of the work and for recommending publication in *Nature Communications*.

From the mechanistic proposal shown in Figure 1D, it appears that the Selectfluor is required to oxidize the ArI to an $ArIF_2$ as described in the report by Shreeve and co-workers (reference 43). However, it remains unclear to me how the intermediate I is formed and why an excess of HF is required for the reaction to proceed. I suspect that the authors assume an initial protonation of the $ArIF_2$ species by the amine HF to yield a cationic $ArIF^+$, which then raises the question of whether the reaction also proceeds with a Brønsted acid other than HF. The authors should explain the role of the amine-HF complex in more detail in the manuscript. In this sense, the authors should also provide information about the nature of the amine-HF complex in the manuscript and not only in the Supporting Information. Cotter, J. L., Andrews, L. J. & Keefer, R. M. The Trifluoroacetic Acid Catalyzed Reaction of Iodobenzene Dichloride with Ethylenic Compounds. *J. Am. Chem. Soc.* **84**, 793-797 (1962).

Author response: We thank the referee for this insightful comment. In the proposed mechanism, the aryl iodonium difluoride is *in situ* generated with aryl iodide as the catalyst, Selectfluor as the terminal oxidant and HF as the fluoride source. On the basis of Hara and Yoneda's seminal study (*Synlett* **1998**, 495-496 (1998)), it was envisaged that activation of the aryl iodonium difluoride with HF occurs by H-bonding, thereby polarizing the F-I-F moiety. This is fully in-line with the report by Cotter et al. that TCA is essential to activate $PhICl_2$ as a chlorinating agent (*J. Am. Chem. Soc.* **84**, 793 (1962) - cited as

reference 46). We have studied the fluxional behaviour of the intermediate by NMR (citation 44: *J. Org. Chem.* **82**, 11792 (2017)), and established that there is fluxional behavior between the ArIF_2 species and the ArIF^+ species in which the reduced Selectfluor[®] core engages with the cation. Engagement of the allene substrate through ligand exchange then affords the intermediate **I**. Therefore, the HF serves a dual role as both fluoride source and Brønsted acid activator, and a large excess is required. This is in-line with previous observations from this lab and many others on optimized fluorination conditions using the I(I)/I(III) manifold (references 34-42 in the manuscript). Following the referee's suggestion, the reaction with HBF_4 was performed and the results are shown in the response to reviewer #1. Regrettably, we did not observe formation of the propargylic fluoride product (by ^{19}F NMR), highlighting the importance (and multiple roles) of the HF source. For more details, we respectfully direct the referee to the response to the question (1) of referee 1. The manuscript has been modified to provide the detailed information of the amine-HF complex. Thank you for this suggestion.

I also suggest that the authors provide an explanation for why alpha-branched alkylallenes (such as commercially available cyclohexyllallene) were not included in the substrate scope.

Author response: This is an excellent suggestion and we thank the reviewer for highlighting this gap in the scope. We exposed cyclohexyllallene to the optimised reaction conditions and were very happy to observe that the desired product was formed in 54% yield (branched to linear ratio 11:1). This result has been added to the manuscript (**Fig. 3**, compound **12**).

Minor: Supporting Information, page 27: Should the sentence: "This compound was prepared in a previous report from our research group." read as "This compound was prepared according to a previous report from our research group."?

Author response: Thank you for this suggestion. The SI has been modified.

• Reviewer #3 (Remarks to the Author):

The manuscript by Gilmour et al. entitled "Regioselective Fluorination of Allenes Enabled by I(I)/I(III) Catalysis" describes a strategy for constructing a variety of secondary and tertiary propargylic fluorides, which are prevalent across the bioactive small molecule spectrum. The main premise of the manuscript is the regioselectivity problem in allene fluorination using Selectfluor being solved using this method. The reaction developed is indeed interesting and displays broad scope. At the end, authors also demonstrate the use of other nucleophiles leading to substituted products. In general, the reaction of Selectfluor with allenes in the presence of nucleophiles has been reported in the literature, leading to different motifs. As such, this method does provide a convenient access to propargylic fluorides but the conceptual novelty is not enough to justify publication in Nature Communications, especially given the absence of a successful enantioselective version (only one example is shown with low enantioselection). While this work is a nice twist to allene-Selectfluor reactivity, it falls short of the expectations of an article to be published in this journal. Addressing the following points may help enhance the manuscript

Author response: We thank referee 3 for their very helpful suggestions on how to further improve the manuscript. Given that the importance of propargylic fluorides in medicinal chemistry and the potency of F to OH bioisosterism, developing efficient methods to synthesize propargylic fluorides is highly desirable. Although the reaction of Selectfluor with allenes in the presence of nucleophiles has been reported in the literature, there is only one example, enabling the formation of propargylic fluorides (*Chem. Commun.* 4113-4115 (2006).). Importantly, this method requires allenylsilanes and not unfunctionalized allenes, as the mechanism requires a 1,2-silyl shift. In this paper, an operationally simple, I(I)/I(III) catalysis-based strategy has been developed to generate various secondary and tertiary propargylic fluorides with good functional group tolerance, and address a long-standing regioselectivity challenge (branched versus linear) in the I(III)-mediated fluoro-functionalization of allenes.

In an attempt to improve the enantioselective process, we have further optimized the reaction conditions and elevated the enantiomeric ratio of **25** to 76.5:23.5 with a novel chiral aryl iodide catalyst (reported by Zhang and co-workers, *ACS Catal.* **13**, 8273-8280 (2023).). This information has been added to the manuscript.

1. Is there an explanation of why the yields were low in the case of aryl allenes (Fig. 3).

Author response: The aryl allenes used in our study were not particularly stable even at -20 °C, and the reactions were conducted at ambient temperatures. To illustrate this, the ¹H NMR of freshly prepared CF₃ substituted aryl allene is shown below:

Following storage of this compound for 12 days at -20 °C, the NMR spectrum is then as follows:

In the case of **S17**, after the specified reaction time of 24 h, 14% starting material was recovered which led to the formation of **17** in lower yield.

2. What is the role of cyclopropyl (**11**) or cinnamoyl motifs (**14**) for the high chemoselectivity?

Author response: Since I(I)/I(III) catalysis-based 1,1-difluorinations of cinnamic acid derivatives (*Science* **353**, 51-54 (2016)) and 1,3-difluorination of cyclopropanes (*J. Am. Chem. Soc.* **139**, 9152–9155 (2017)) have been reported by Jacobsen and coworkers, allene substrates containing these motifs (cyclopropane (**S11**) and cinnamoyl motifs (**S15**)) were prepared and exposed to the optimized conditions to explore the chemoselectivity of the reaction. Pleasingly, the transformation was found to be chemoselective for the allene versus cyclopropanes and cinnamoyl motifs. We hope that this addresses the question.

3. In the case of tertiary propargylic fluorides, authors have shown limited scope. The late-stage functionalization of important bioactive molecules should also be included, as shown in Fig. 3 (12-14, 21).

Author response: Following the referee's suggestion, we have explored the reaction of 1,1-disubstituted allene derived from *D*-menthol. This led to the formation of the expected tertiary propargylic fluoride in 73% yield. This result has been added to the manuscript (**Fig. 4**, compound **31**). We thank the reviewer once again for this helpful suggestion.

4. Do the authors observe the 1,2 difluorination products (like **37**) in other cases also? If not, what is the rationale for that?

Author response: This is not a general phenomenon and in other cases no 1,2-difluorinated products were observed. A possible rationale for this observation with **38** is that a different mechanism is operational as is shown below. The α -fluorinated allene would undergo 1,3-fluorine migration, leading to the formation of fluorinated diene, which then engages in an I(I)/I(III) catalysis cycle, enabling a regioselective 1,4-difluorination to generate the desired *multi*-fluorinated product. This is a transformation that is currently under investigation in our labs. It is pertinent to note that Gololobov and coworkers also observed the 1,3-fluorine migration in their study (*Russ Chem Bull* **55**, 1309–1310

(2006)), and the catalytic regioselective 1,4-difluorination of dienes was recently reported by our group (*Angew. Chem., Int. Ed.* **62**, e202309789 (2023)).

6. A detailed mechanistic analysis, including computational studies.

Author response: Following the instructions from the editor, a computational investigation has not been conducted.

REVIEWERS' COMMENTS

Reviewer #1 (Remarks to the Author):

The authors have addressed the issues well. This referee recommends acceptance of this manuscript.

Reviewer #2 (Remarks to the Author):

The authors have thoroughly addressed the points I previously mentioned. I am pleased that the substrate scope has now been extended by the addition of cyclohexylallene. Although the enantioselectivity of the fluorination of S25 with a modified catalyst was only slightly improved, I think it is acceptable to include in such a comprehensive manuscript a preliminary experiment that shows that an enantioselective reaction is possible in principle and that the structure of the chiral catalyst influences the ee.

In summary, I recommend that the revised manuscript be accepted for publication in Nature Communications.

Reviewer #3 (Remarks to the Author):

The manuscript was reviewed earlier and was sent for revisions. The authors have performed the experiments suggested by the reviewers and also answered many queries satisfactorily. I recommend this manuscript for publication and congratulate the authors for developing a nice piece of chemistry!